# Separation of Water in Diesel Using Filter Media Containing Kapok Fibers

**DOI:** 10.3390/ma13112667

**Published:** 2020-06-11

**Authors:** Qiang Song, Jian Kang, Min Tang, Yun Liang

**Affiliations:** 1State Key Laboratory of Pulp and Paper Engineering, South China University of Technology, 381 Wushan Rd., Guangzhou 510640, China; s013411143@163.com (Q.S.); liangyun@scut.edu.cn (Y.L.); 2State Key Laboratory of NBC Protection for Civilian, 35 Huayuan Rd. N., Beijing 100191, China

**Keywords:** filter media, water separation, diesel, kapok fiber

## Abstract

Traditional water-repellent filter media for water separation in diesel fail to meet requirements due to the high content of surfactants in low sulfur diesel and ULSD (ultra low sulfur diesel). To improve the water separation performance of filter media, a novel dual-layer filter medium was prepared by hydrophilic fibers (glass microfibers) and hydrophobic fibers (kapok fibers and bi-component PET fibers). The results showed that the separation efficiency of a filter medium (sample #2) with the upstream layer containing 20 wt% kapok fibers was 89.5%, which was higher than that of filter samples with the upstream layer containing 0 wt%, 40 wt%, 60 wt% and 80 wt% kapok fibers. When the interfacial tension (IFT) of water in diesel was 21 mN/m, 17 mN/m and 13 mN/m, the separation efficiency of filter sample #2 was 99.5%, 89.5% and 30.5%, respectively, which was 23.9%, 57.4% and 17.8% higher than that of the commercial water-repellent filter samples composed of a polybutylene terephthalate (PBT) fiber layer and cellulose fiber layer.

## 1. Introduction

Water in diesel fuel accelerates the deterioration of diesel and reduces the combustion efficiency, which caused an increase in engine emissions of hydrocarbons and carbon monoxide [1]. The water and the deteriorated diesel can cause corrosion and blockage of engine components [2]. Therefore, it is important to remove water from diesel for engine protection.

The diesel/water separation is accomplished by a filter, in which the diesel flows through the filter media. The water separation performance of filter media is determined by the emulsion properties (water droplet size, water concentration, interfacial tension (IFT), etc.), filter media properties (pore size, wettability, multi-layer structure, etc.) and the operating conditions (face velocity, temperature, etc.). Recently, low sulfur diesel (sulfur < 500 ppm) and ultra low sulfur diesel (ULSD, sulfur < 15 ppm) have been widely used in the industry [3]. The content of surfactant is higher compared with traditional diesel for better lubricity, which leads to a decrease of water droplet size in diesel and an increase of the stability of diesel/water emulsions [4]. It is difficult for water droplets with smaller sizes to attach to filter media and coalesce into large water droplets for settlement by gravity [5]. In addition, the wettability of filter media, a decisive factor affecting the coalescence of water droplets, changes because the surfactants in diesel adsorb on the surface of filter media [6]. For engines, the application of HPCR (high pressure common rail) fuel systems can also lead to smaller water droplets since the shear speed of the centrifugal pump is higher [7]. Therefore, the filter media for traditional diesel engines cannot meet the requirement of diesel/water separation for low sulfur diesel and ULSD.

The filter media with hydrophobic surface tends to resist the adsorption of surfactants on the filter surface [8]. The commercial filter media made by hydrophobic polybutylene terephthalate (PBT) fibers showed a low separation efficiency when it was tested according to the SAE J1488 standard [9], which indicated that filter media made by hydrophobic fibers cannot solve the current problem of diesel/water separation. In addition to hydrophobic separation media, there were studies on coalescence media and modified PBT filter media [10]. Hamidreza Arouni et al. studied that the optimal wettability of modified PBT filter samples was different for diesel containing different surfactants to obtain the highest coalescence efficiency [11]. Kulkarni et al. studied that the separation performance of filter media prepared by a mixture of hydrophilic/hydrophobic fibers was better than that of filter media made by pure hydrophilic fibers or hydrophobic fibers [12]. The separation performance of single layer filter media prepared by a mixture of hydrophilic/hydrophobic fibers was better than that of dual-layer filter media composed of a hydrophilic fiber layer and a hydrophobic fiber layer [13]. Gadhave et al. studied that the coalescence performance of filter samples composed of double layers with different wettability of each layer was better than that with same wettability of each layer [14]. However, filter media made by a mixture of hydrophilic and hydrophobic fibers was difficult to be prepared since the hydrophobic fibers in water tended to form flocculates.

Kapok fiber is a kind of hydrophobic and oleophilic natural fiber with a large lumen structure. Many studies showed that kapok fibers were applied in separating oil from water due to its high oil adsorption ratio [15,16]. Chaiarrekij et al. proposed that kapok fibers had a potential for pulping and papermaking [17], and the water contact angle of paper made by 100% kapok fiber was 70° or less. The filter paper prepared by mixing kapok fibers with softwood fibers [18] or glass microfibers [19] was applied in filtrating oil aerosols. The results demonstrated that kapok fibers could improve the oil aerosol holding capacity. In addition, it could be seen that the water contact angle of paper made by pre-treated kapok fibers was lower than paper made by untreated kapok fibers due to the destruction of the waxy surface of the kapok fibers by NaOH [20]. However, kapok fibers pre-treated by NaOH were difficult to be applied in diesel/water separation due to decrease of hydrophobicity.

Currently, the application of kapok fibers in diesel/water separation has not been reported. In this study, kapok fibers were used to design a novel filter medium with a dual-layer structure composed of a hydrophilic/hydrophobic fiber layer and a hydrophobic fiber layer. The pore size of filter media was a key factor affecting the filtration performance according to the fluid dynamics [21]. Moreover, the dynamic wetting performance of water droplets on fibers with different water wettability was discussed. Compared with hydrophobic synthetic fibers, kapok fibers were renewable, biodegradable and easy to be dispersed in water by a new method proposed by this study while maintaining its hydrophobic surface. Therefore, this study helps the development of new filter media for diesel/water separation.

## 2. Materials and Methods

### 2.1. Fibers and Chemicals

The kapok fibers were produced in Indonesia. The bi-component PET fiber was provided by Teijin Limited (TJ04CN, Tokyo, Japan). The glass microfiber was supplied by Zaisheng Technology Co., Ltd. (#253-39, Chongqing, China). Cetyltrimethylammonium bromide (CAS: 57-09-0) was offered by Beijing Coupling Technology Co., Ltd. (Beijing, China). Diesel (0#) was purchased from a gas station in Guangzhou, China. Monoolein (CAS: 25496-72-4) was provided by Sino-Japanese Synthetic Chemical Co., Ltd. (Taipei, China). Active clay was purchased from Huangshan Baiyue Activated White Earth Co., Ltd. (Anhui, China). Commercial filter samples composed of a polybutylene terephthalate (PBT) fiber layer and the cellulose fiber layer were provided by Fibrway Co., Ltd. (Guangzhou, China).

### 2.2. Filter Media Preparation

As shown in Figure 1, the filter medium was composed of an upstream layer and a downstream layer. The upstream layer was mixed with hydrophilic fibers (glass microfibers) and hydrophobic fibers (kapok fibers and bi-component PET fibers), which was for coalescence of fine water droplets. The downstream layer prepared by hydrophobic fibers (kapok fibers and bi-component PET fibers) was for the drainage of large water droplets. It also reduced the pressure drop across the filter media due to the weak adhesion with water droplets [22]. The diameter of different fibers is shown in Table 1. It can be seen that the average diameter of hydrophilic fibers was finer than that of hydrophobic fibers, which could contribute to reduction of pore size of filter media [23]. In Table 2, the percentage of kapok fibers in the upstream layer was 0 wt%, 20 wt%, 40 wt%, 60 wt% and 80 wt%, respectively, and the downstream layer of all filter media was made by 80 wt% kapok fibers and 20 wt% bi-component PET fibers.

Different fibers were weighed according to the fiber composition, as seen in Table 2. First kapok fibers were dispersed in 1000 mL water by high shear disintegrator (model: HR210, Philips, Amsterdam, Netherlands) for 8 min. Then glass microfibers and bi-component PET fibers were added to the slurry of kapok fibers for 2 min disintegration. The disintegrated slurry was diluted to 0.06 wt% in the sheet former (model: RK3AKWT, PTI GmbH, Laakirchen, Austria), and cetyltrimethylammonium bromide was added with its concentration in the slurry of 32 ppm by weight. In the sheet former, the wet sheet of upstream layer was formed after water drainage. The preparation process of downstream layer was the same as upstream layer. Finally, the two layers were combined and dried in a 120 ℃ rotary dryer. The skin of bi-component PET fibers melted during drying, so that the filter media were reinforced. Thus, the media could keep the integrity in the water separation test. Five samples with basic weight of 200 ± 5 g/m^2^ were prepared for each kind of filter medium, as seen in Table 2.

### 2.3. Characterization of Fibers and Filter Media

The contact angle of water droplet on fiber was an essential parameter to be characterized for water separation. Before measuring the contact angle, the filter papers were immersed in the diesel with IFT of 21 mN/m, 17 mN/m and 13 mN/m. IFT was adjusted by adding different content of monoolien. IFT was measured by a force tensiometer (QBZY-2, Fangrui Instrument Co., Ltd., Shanghai, China) at an interfacial age of 60 s according to ASTM D971-12 [24]. Then, the photos of water droplets on paper samples in diesel were captured by a contact angle surface analyzer (OCA25, Eastern-Dataphy Instruments Co., Ltd., Beijing, China). The contact angle of water droplet on fiber could demonstrate the water wettability of the fiber.

Several physical properties of filter media concerning water separation performance were measured. The pore size of filter media was measured by capillary flow porometer (model: CFP-1100-A, Porous Materials Inc., Ithaca, NY, USA). The thickness of samples was measured by a thickness tester (model: YG142, Ningbo Textile Instrument, Ningbo, China) with pressure of 0.5 kPa applied on the samples. The air permeability of filter media was tested by an air permeability tester (model: FX3000, Textest Instruments AG, Schwerzenbach, Switzerland). The morphology of filter media was analyzed by SEM (model: G2Pro Y, Phenom-World, Eindhoven, Netherlands).

### 2.4. Diesel/Water Separation Performance

The diesel was treated by adding monoolein or clay to achieve different IFT. When IFT reached 21 mN/m, 17 mN/m or 13 mN/m (according to the recommendation of SAE J1488-2010 [9] and ISO 16332-2018 [25]), respectively, water was pumped into an emulsion sump to prepare a water/diesel emulsion (with water concentration of 2500 ppm in the diesel) by a high speed disperser (model: 78-1, IKA, Staufen, Germany) for 30 min. The content of water in the emulsion was tested by Karl Fischer titration (model: C20, METTLER TOLEDO, Greifensee, Switzerland) according to ASTM D6304 [26]. The size distribution of water droplets was measured by a laser diffraction particle size analyzer (model: Mastersizer 3000, Malvern Panalytical, Malvern, England).

The diesel/water separation test rig was designed based on SAE J1488 [9] to characterize the water separation performance of filter media. The schematic diagram of the test rig is shown in Figure 2. The fuel sump was used for fuel storage and used fuel collection, and the emulsion sump was used to store treated fuel for the test. First, the diesel/water emulsion was generated by above method. The volume concentration of dissolved water in diesel was determined according to Appendix B in SAE J1488 [9]. Then, the emulsion was pumped to a filter holder at a face velocity of 2 cm/min. The concentration of water in the upstream and downstream was measured every 10 min by the above-mentioned Karl Fischer titration. The experiment was terminated when a steady state was achieved for at least 30 min. The water separation efficiency of filter media was calculated by the following Equations ((1) and (2)) according to the concentration by volume of water in the upstream (2500 ppm) and downstream:(1)$$C_{down}=\frac{\sum_{i}^{n}\left(C_{down,i}-C_{dissolve}\right)}{n},$$
(2)$$E=(1-\frac{C_{down}}{2500})\times 100\%,$$
where *C_down_* was the average concentration by volume of water in downstream, ppm; *C_down,i_* was the concentration by volume of water in downstream at steady state, ppm; *C_dissolve_* was the concentration by volume of dissolve water in diesel, ppm; *n* was the number of times tested at steady state; *E* was the separation efficiency. Pressure drop of filter media was recorded by a differential pressure gauge.

To study the effect of the content of kapok fibers on separation performance, the efficiency and pressure drop of filter media were tested when IFT was 17 mN/m. The filter medium with highest efficiency was selected to compare with the commercial filter medium (Fibrway, Guangzhou, China) when IFT was 21 mN/m, 17 mN/m and 13 mN/m, respectively.

## 3. Results and Discussion

### 3.1. The Wettability of Fibers Under Different IFT

The wettability of fibers significantly affected interaction between water droplets and fibers. The water contact angle on paper samples made by hydrophobic fibers is shown in Figure 3. It can be seen that when IFT decreased from 21 mN/m to 13 mN/m, the water contact angle on all paper samples decreased slightly. The water contact angle of kapok fibers was higher than that of PBT fibers and PET fibers under the same IFT, which demonstrated that kapok fibers had excellent hydrophobic properties. Furthermore, it can be seen that the shape of water changed from sphere to ellipsoid as IFT decreased from 21 mN/m to 13 mN/m, indicating that the concentration of surfactant in diesel had a significant impact on stability of water droplets due to the adsorption of surfactant on the interface of diesel/water [6]. As a result, the coalescence of water droplets in diesel became more difficult under low IFT, which could lower the separation efficiency of the filter media.

The water wettability of samples made by glass microfibers is shown in Figure 4. It can be seen that for the starting moment (0 s), the water contact angle on samples in diesel increased from 102° to 138° as surfactants caused IFT to decrease from 21 mN/m to 13 mN/m. When IFT was 21 mN/m and 17mN/m, the water was adsorbed by samples in 19 s and 46 s, respectively. The water contact angle in diesel with IFT of 13 mN/m was more stable. The results indicated that as surfactants caused IFT to decrease from 17 mN/m to 13 mN/m, the surface property of glass microfiber samples was turned from wetting to non-wetting, leading to an increase in the probability of water droplet detachment on glass microfiber.

### 3.2. The Physical Properties and Morphology of Filter Media Containing Kapok Fibers

The basic properties of filter media with different content of kapok fibers are shown in Table 3. As shown in Figure 5, the mean pore size of upstream layer (9.6 μm–11.3 μm) was slightly larger than the median size of water droplets (8.5 μm, as discussed in Section 3.3), which promoted the coalescence of water droplets on filter media [21]. When water droplets contacted the downstream layer (with mean pore size of 19.1 μm to 19.8 μm), droplets smaller than the pore size would coalesce by impacting or penetrate through the downstream layer. The water droplets larger than pore size would be held by hydrophobic fibers, or penetrate through the downstream layer by deformation, leading to an increase of residual time in the filter media and a higher probability of coalescing.

The maximum pore size gradually increased from 18.7 μm to 29.7 μm as the content of kapok fibers increased. Larger size pore meant that more water droplets could penetrate through the filter media. The thickness of filter media gradually increased from 0.98 mm (0 wt% kapok) to 1.48 mm (60 wt% kapok) and then decreased to 1.15 mm (80 wt% kapok). The air permeability decreased from 92 mm/s to 45 mm/s as the content of kapok fibers increased from 0 wt% to 80 wt%.

The morphology of the upstream layer containing different contents of kapok fibers is shown in Figure 6. In Figure 6a, glass microfibers intertwined with PET fibers in filter media. When the mass percentage of kapok fibers was higher than 8 wt%, the volume percentage of kapok fibers was larger than that of glass microfibers, which could be calculated from the density of kapok fibers (0.29 g/m^3^) and glass microfibers (2.55 g/m^3^). In Figure 6b, the upstream layer with 20 wt% kapok fibers formed the fibrous network. Glass microfibers bridged across kapok fibers, which could effectively reduce the pore size of filter media. When the content of kapok fibers reached the range of 40 wt% to 60 wt%, the distance of glass microfibers was increased. When the content of kapok fibers reached 80 wt% (Figure 6e), the distance of fibers was further reduced without intertwining with glass microfibers.

### 3.3. Evaluation of Water Separation Performance for Media with Different Kapok Fiber Content

#### 3.3.1. Effects of Surfactant on The Diesel/Water Emulsion Properties

As shown in Figure 7, IFT decreased with the increase of monoolein. Monoolein can affect the molecular interaction in the water/diesel interface [27]. Monoolein was added to achieve concentrations of 115 ppm, 180 ppm and 263 ppm by volume in the emulsion, respectively, which led to IFT of 21 mN/m, 17 mN/m and 13 mN/m, respectively. As seen in Figure 8, as IFT decreased from 21 mN/m to 17 mN/m, the size distribution of water droplets was narrowed. D90 (the intercepted droplet diameter at 90% by volume) decreased from 25.5 μm to 17.6 μm, and D50 (the median diameter of droplets) decreased from 12.9 μm to 8.5 μm. In the process of forming water droplets by shear force, the surfactants in diesel were adsorbed in the diesel/water interface [28], leading to high stability of water droplets. The coalescence of water droplets became more difficult in the diesel/water emulsion [29]. Therefore, the water droplets produced by shearing tended to keep the original size. When the IFT decreased to 13 mN/m, D90 and D50 were 18.0 μm and 7.3 μm, respectively, which was close to that of the emulsion under IFT of 17 mN/m. The water droplets with large sizes tended to contact with fibers by deformation. As the droplet size decreased, the water droplets tended to keep the spherical shape on fibers with slight deformation, resulting in lower probability of water droplet coalescence, since deformation of droplets hindered the coalescence [30]. In addition, contact time among smaller droplets was shorter, and it was more likely for small droplet to approach each other. Therefore, the separation efficiency of filter media decreased with smaller water droplet size.

#### 3.3.2. Effects of The Content of Kapok Fibers on Water Separation Performance

The separation efficiency and pressure drop of the filter media with different kapok fiber content under IFT of 17 mN/m are shown in Figure 9. It can be seen that as the content of kapok fibers in the upstream layer increased from 0 wt% to 20 wt%, the separation efficiency of filter media increased from 82.2% to 89.5% and the pressure drop increased from 1.5 kPa to 1.9 kPa. When the content of kapok fibers in the upstream layer increased from 20 wt% to 40 wt%, the separation efficiency decreased rapidly to 43.3%. When the content of kapok fibers in the upstream layer increased from 40 wt% to 80 wt%, the separation efficiency rapidly increased to 65.9%, and the pressure drop decreased from 1.5 kPa to 0.9 kPa.

The separation performance of filter media with the same downstream layer depended on the fiber wettability and structure of the upstream layer. When diesel/water emulsion with IFT of 17 mN/m passed through the filter media, the water droplets impacting on hydrophilic fibers (glass microfibers) tended to be captured by fibers and then coalesced with other water droplets. The water droplets impacting on hydrophobic fibers (kapok fibers and PET fibers) tended to bounce back into the emulsion, which improved the concentration of water droplets in the domain of glass microfibers. Thus, the probability of capturing water droplets by glass microfibers was increased. Therefore, the efficiency of the filter medium was improved from 82.2% to 89.5% as the content of kapok fibers in the upstream layer increased from 0 wt% to 20 wt%. When the percentage of kapok fibers in the upstream layer increased to 40 wt%, the amount of glass microfibers decreased, which led to lower separation efficiency. When the percentage of kapok fibers in the upstream layer increased to 80 wt%, the total surface area of fibers increased, and the reduction of glass microfibers was compensated. The probability of water droplets impacting on fibers was largely increased.

#### 3.3.3. Effects of IFT on Water Separation Performance

Figure 10 shows the morphology of the commercial filter medium. It can be seen that the commercial filter medium was composed of an upstream layer (made by PBT fibers) and a downstream layer (made by softwood and hardwood fibers). The basic properties of commercial filter medium are shown in Table 4. It can be seen that the basic weight of the commercial sample (224 g/m^2^) was close to that of filter medium #2 (200 g/m^2^). The mean pore size of the commercial sample was 7.1 μm, which was also close to that of filter medium #2 (7.6 μm).

The separation performance of commercial sample was compared with that of filter medium #2. As seen in Figure 11a, when IFT was 21 mN/m, 17 mN/m and 13 mN/m, the separation efficiency of filter medium #2 was 99.5%, 89.5% and 30.5%, which was 23.9%, 57.4% and 17.8% higher than that of the commercial sample, respectively. The results indicated that the prepared filter medium had a better water separation performance than commercial media. The reason was that the hydrophilic fibers among the hydrophobic fibers promoted the coalescence of water droplets. As seen in Figure 11b, the pressure drops under saturation conditions of filter medium #2 varied from 1.9 kPa to 2.5 kPa, which was relatively stable compared with that of the commercial sample (from 3.2 kPa to 1.2 kPa).

## 4. Conclusions

To improve the performance of filter media in removing water from low sulfur diesel or ULSD, a novel dual-layer filter medium was prepared by incorporating hydrophilic fibers and hydrophobic kapok fibers. The water separation performance of the novel filter medium was evaluated. The results showed that the filter medium with an upstream layer containing 20 wt% kapok fibers possessed maximum separation efficiency (89.5%) under IFT of 17 mN/m. Under different IFT, the efficiency of dual-layer filter media containing 20 wt% kapok fibers was significantly higher than that of the commercial media. Therefore, according to the dynamics and interface properties of water droplets, combining kapok fibers with hydrophilic fibers in filter media design can effectively improve the separation performance. In the future, more studies will be conducted on the dynamic processes of water separation inside the kapok fiber media to optimize filter media containing kapok fibers.

## Figures and Tables

**Figure 1 materials-13-02667-f001:**
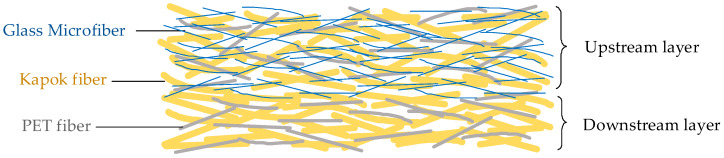
The dual-layer structure of filter media containing kapok fibers.

**Figure 2 materials-13-02667-f002:**
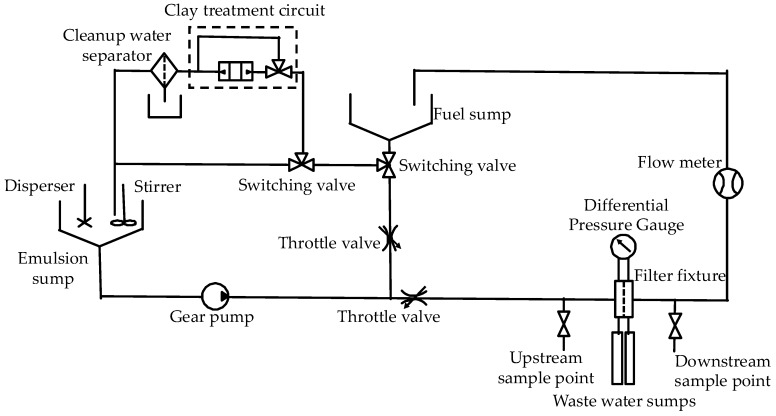
The schematic diagram of the diesel/water separation test rig for filter media.

**Figure 3 materials-13-02667-f003:**
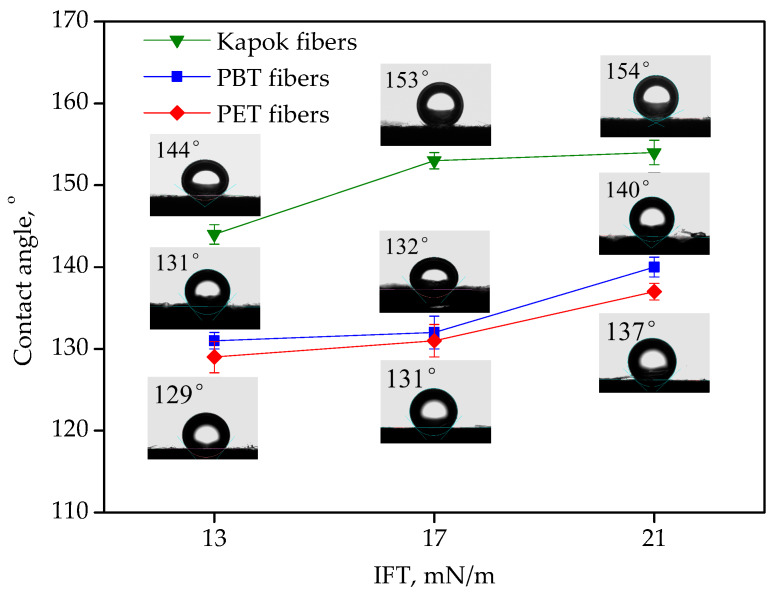
The contact angle of fibers under different interfacial tension (IFT).

**Figure 4 materials-13-02667-f004:**
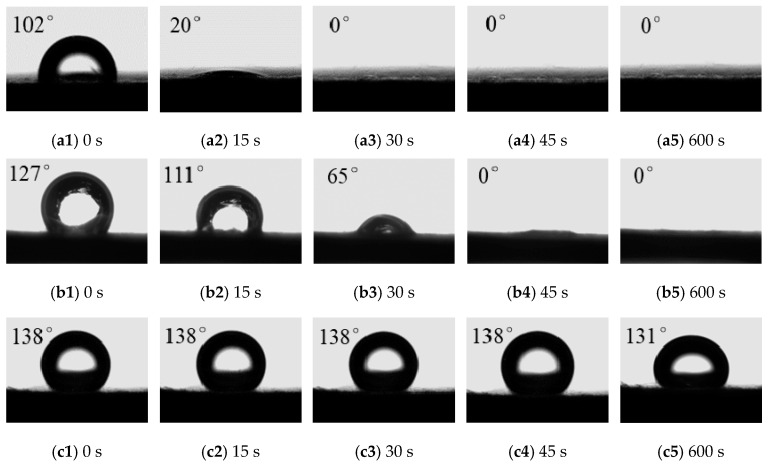
The water contact angle on samples made by glass microfibers: (**a**) IFT = 21 mN/m; (**b**) IFT = 17 mN/m; (**c**) IFT = 13 mN/m.

**Figure 5 materials-13-02667-f005:**
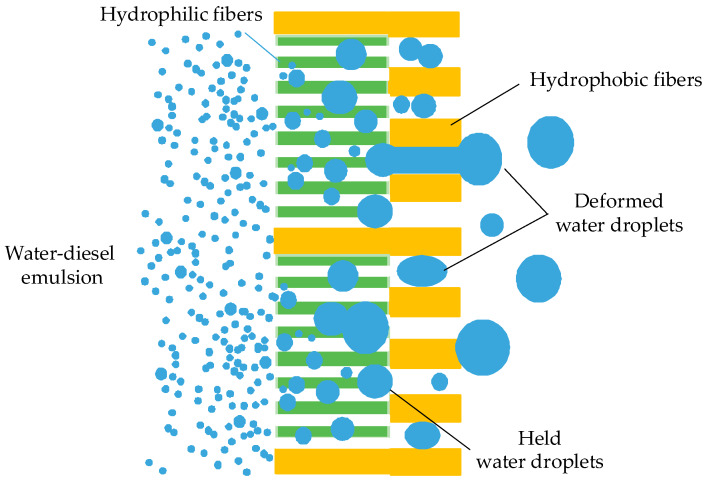
The diagram of coalescence of filter media prepared by hydrophilic/hydrophobic fibers.

**Figure 6 materials-13-02667-f006:**
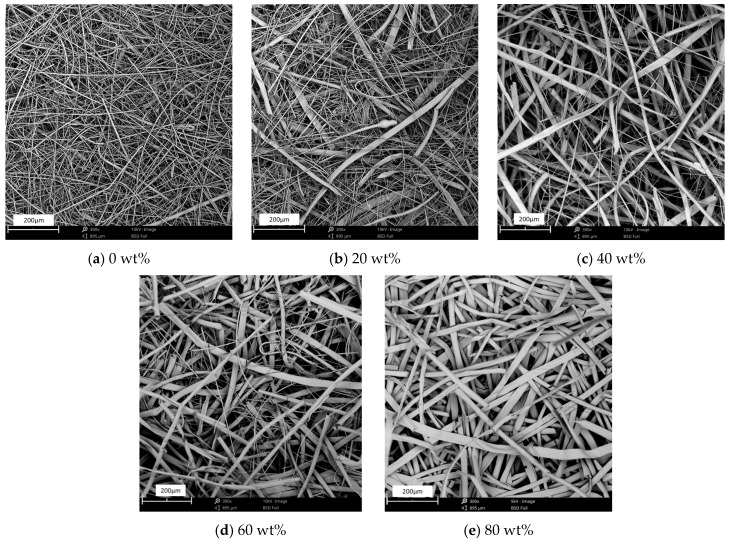
SEM images of filter media with different kapok fiber content.

**Figure 7 materials-13-02667-f007:**
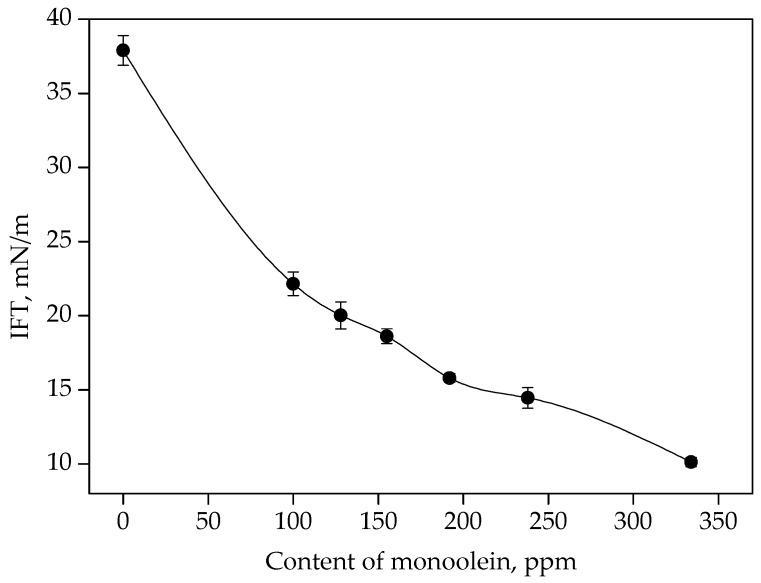
IFT under different contents of monoolein.

**Figure 8 materials-13-02667-f008:**
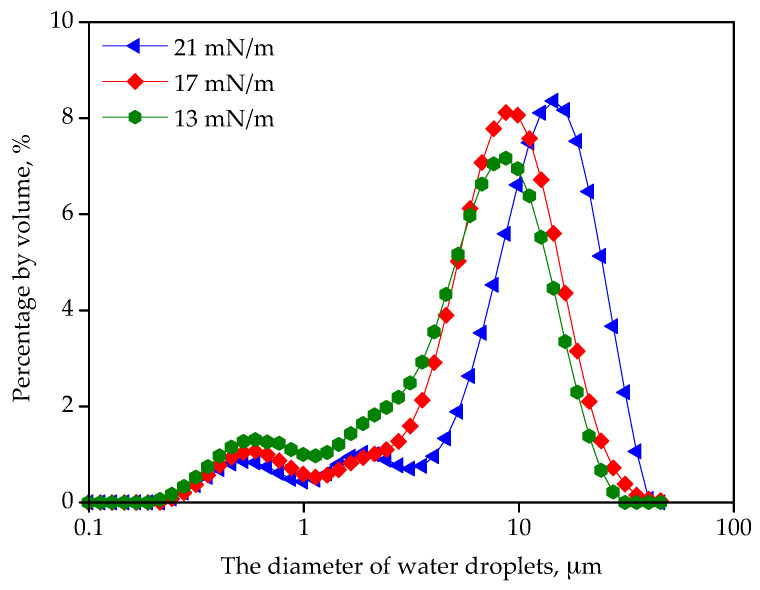
The size distribution of water droplets in diesel under different IFT.

**Figure 9 materials-13-02667-f009:**
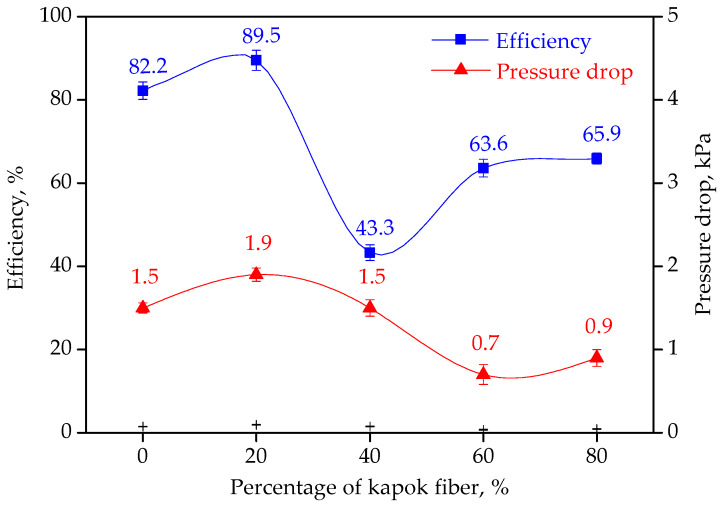
Effect of kapok fiber content on separation efficiency and pressure drop of filter media.

**Figure 10 materials-13-02667-f010:**
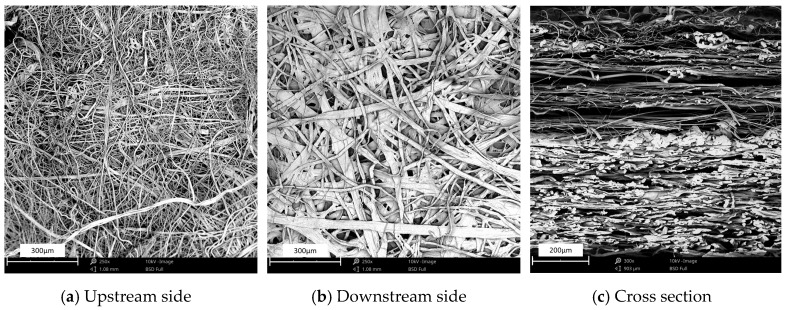
SEM images of commercial filter media.

**Figure 11 materials-13-02667-f011:**
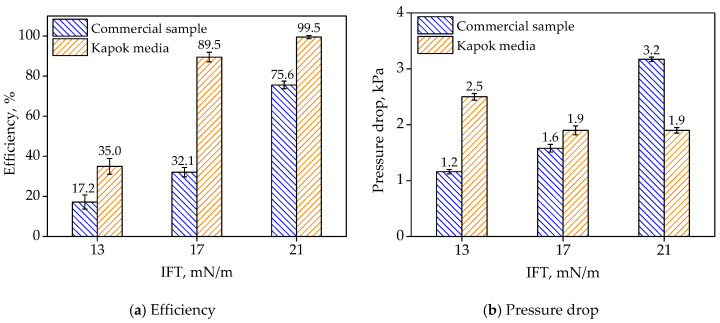
The diesel/water separation performance of filter media under different IFT.

**Table 1 materials-13-02667-t001:** The diameter of different fibers.

Fiber Parameter	Glass Microfiber	Kapok Fiber	PET Fiber
Diameter Range (μm)	0.3–9.0	12.0–35.0	12.5–15.2
The Average Diameter (μm)	2.2	23.0	13.1

**Table 2 materials-13-02667-t002:** The percentage of different fibers in filter media.

Layer	Basic Weight (g/m^2^)	Fiber Type	Percentage of Fibers (wt%)				
			1#	2#	3#	4#	5#
Upstream Layer	150	Kapok Fiber	0	20	40	60	80
		Glass Microfiber	80	60	40	20	0
		PET Fiber	20	20	20	20	20
Downstream Layer	50	Kapok Fiber	80	80	80	80	80
		PET Fiber	20	20	20	20	20

**Table 3 materials-13-02667-t003:** The basic properties of filter media with different content of kapok fibers.

Basic Properties		#1	#2	#3	#4	#5
Percentage of Kapok Fibers (wt%)		0	20	40	60	80
Mean Pore Size (μm)	Upstream Layer	7.5 ± 0.1	8.3 ± 0.1	7.3 ± 0.2	7.6 ± 0.1	11.3 ± 0.3
	Downstream Layer	19.4 ± 0.2	19.7 ± 0.3	19.8 ± 0.4	19.1 ± 0.3	19.6 ± 0.5
	Composed Filter	7.0 ± 0.1	7.6 ± 0.2	6.6 ± 0.3	6.6 ± 0.3	7.8 ± 0.4
Maximum Pore Size (μm)		18.7 ± 0.5	20.1 ± 0.7	24.2 ± 0.6	27.1 ± 0.4	29.7 ± 0.9
Thickness (mm)		0.98 ± 0.03	1.04 ± 0.05	1.08 ± 0.04	1.48 ± 0.02	1.15 ± 0.05
Air Permeability (mm/s)		92 ± 1	89 ± 2	63 ± 1	47 ± 1	45 ± 1

**Table 4 materials-13-02667-t004:** The performance of commercial filter medium and sample #2.

Filter Media	Basic Weight (g/m^2^)	Thickness (mm)	Air Permeability (mm/s)	Mean Pore Size (μm)	Maximum Pore Size (μm)
Commercial Filter Sample	224 ± 4	0.92	41	7.1	20.3
Sample #2	200 ± 2	1.04 ± 0.05	89 ± 2	7.6 ± 0.2	20.1 ± 0.7

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
