# Peer review of "Separation of Water in Diesel Using Filter Media Containing Kapok Fibers"

_materials, 2020, doi:10.3390/ma13112667_

Round 1
Reviewer 1 Report
All remarks are included in the attached file.

Author Response
The presented paper concerns an important and up-to-date topic of diesel fuel dewatering. Although the diesel market is shrinking with regards to small cars, the diesel motor driven vehicles will still possess a significant share on the market of heavy duty transport. In recent years the water removal became a big challenge for the filtration system due to change of fuel composition. However, the biggest issues arise from addition of biocomponents, which is imposed by legislation to promote renewable energy sources in transport.
Authors investigate the process of the diesel fuel dewatering by preparing the filter media containing the blend of kapok fibers. In first part Authors refer to traditional hydrophobic filter elements – water separators – which fail in operation with ULSD. They however do not point out that operation of coalescence media is completely different process (flow through of both phases, unlike in typical hydrophobic water separators, where only the fuel passes through the media).
Ans: We would like to thank the reviewer for the comment. The coalescence media was mentioned in the first part.
The first section is a review on the water separation performance for different filter media. The Authors refer to a commercial product made of PBT polyester, which performed badly. However, in recent literature (doi 10.1016/j.seppur.2019.116254 published in 2020) you can find a relevant research on PBT modified coalescence filter media, which perform quite well as a coalescers (depth media). It is worth to include this in the review.
Ans: We would like to thank the reviewer for the comment. We have included the recent literature on PBT modified filter media in the review. This literature is very helpful.
The experimental setup and materials used are clearly describer. The results are well presented, however they are not sufficiently well discussed and analyzed – a better interpretation of observed tendencies is strongly recommended (this refers to lines 232-254), where the maximum and then the minimum of the sep. efficiency vs. content of kapok fibers was observed (Fig. 9). The conclusion “When the percentage of kapok fiber in upstream layer increased to 80%, the total volume of fibers was improved (should be increased; in fact not the volume, but specific surface area is an important parameter for the filtration process as it is related to the area for deposition). The probability of impacting of water droplets on fibers was largely increased, which led to the efficiency of filter medium increased to 65.9%.” is questionable – this can be just an effect of dense packing and not the nature of fiber types (the permeability was reduced).
Ans: As also mentioned by other reviewers, this discussion was questionable. We think that with higher kapok fiber content, it was obvious the efficiency would decrease due to lower content of glass microfiber. Since the density of glass fiber was more than 8 times higher than kapok fiber, the total number of kapok fiber in media increased. The total surface area also increased, which accounted for higher droplet attachment. It compensated the reduction of glass microfiber, and the efficiency increased when kapok content was higher than 60%. We modified the explanation in the manuscript.
Lines 178-179: where this conclusion comes from that the probability of droplet attachment is related to fiber wettability? No doubt they detach more easily from the hydrophobic fibers, but does it affect attachment? The Figure 4 gives some insight how the wettability is changed, but not how the efficiency of attachment is affected by the wettability of the surface.
Ans: The reviewer’s comment was correct. The droplet attachment was more likely to be linked with fiber diameter rather than fiber wettability. We have changed the statement to “leading to a decrease in the probability of water droplets detachment on glass microfiber”.
Line 172: Description of Figure 3 is not correct – this is not a size distribution of droplets.In multiple places Authors use term “kapok fiber was” – it should be used in plural form, i.e. “kapok fibers were” (for example line 56, 58, 60, 67, 72 etc.).
Ans: Sorry for the error. The caption was revised, and “kapok fiber was” was changed to “kapok fibers were”.
Line 65: “hydrophobic performance” - what is that?? English
Ans: “hydrophobic performance” was changed to “hydrophobicity”.
Lines 69-70: what means that “pore size was delicately controlled”?
Ans: This sentence was deleted.
Line 118: “..process of water separation test.” ??? English
Ans: Sorry for the confusion. It was changed to “in the water separation test”.
Figure 2: Please at least comment the need for two water collecting sumps in the filter housing.
Ans: The sump in the right was for fuel storage, and the one in the left was for emulsion treatment. The statement about the two pumps was added in the manuscript.
Lines 183-190: written in bold.
Ans: We didn’t understand the meaning of this comment. What does it mean by “written in bold”?
Line 198-199: If the air permeability was reduced during the kapok fibers were added, it is difficult to draw conclusion on their effect on the separation performance – this can be effect of the structure and not properties (wettability) only. I am aware that in practice it is difficult to isolate both effects (change only one parameter), but when you significantly change two parameters, it is difficult to draw a justified conclusion.
Ans: Thanks for the insight of the reviewer. Indeed it is difficult to isolate the impact of fibrous structure and fiber wettability. What we can find is that for sample #2, we can obtain both higher separation efficiency and air permeability. In the future, we will use plant fiber with close fiber diameter compared with kapok fiber to further verify the impact of fiber wettability.
Line 219: “D90 and D50 was 18.0 μm and 7.3 μm, respectively” – were
Ans: Sorry for the mistake. “was” has been changed to “were”.
Line 220: “In this case, the adsorption capacity of surfactant in the diesel/water interface reached saturation when the IFT was 17 mN/m and 13 mN/m [21].” I am not sure what Authors understand as “saturation” – the CMC is for sure not reached (see decrease below 13 mN/m in Figure 7).
Ans: Thanks for the comment. This statement was incorrect. We have deleted it.
Lines 222-225: “As the droplet size decreased, the water droplets tended to keep the spherical shape on fiber with slight deformation, resulting in lower probability of water droplets coalescence [22]. Therefore, the separation efficiency of filter media decreased with smaller water droplets size.” – the stiffness of the interface is in general beneficial for the coalescence, which can be easily explained by assumption of the film drainage models for binary coalescence (and the deformation of the interface hinders the coalescence). But when the droplet size is reduced then the contact time is shorter and what is even more important, the repulsive interactions play more important role, when the droplets approach each other. The provided description should be rearranged (perhaps making use of above information).
Ans: Thanks for the information of the reviewer. We have added your opinion into the manuscript. The discussion in the manuscript was “As the droplet size decreased, the water droplets tended to keep the spherical shape on fiber with slight deformation, resulting in lower probability of water droplets coalescence since deformation of droplet hindered the coalescence. In addition, contact time among smaller droplets was shorter, and it was likely for small droplet to approach each other. Therefore, the separation efficiency of filter media decreased with smaller water droplets size.”
Line 238: “the pressure drop decreased from 0.7 kPa to 0.9 kPa.” – this is an increase (maybe it should be from 1.5 to 0.9?).
Ans: Sorry for the mistake. It was changed to “1.5 kPa to 0.9 kPa”.
Figure 9: The curve fit in the graph applied in the software causes an artificial maximum of the line (on the left from the experimental point); and the same on the right from the 43.3 value.
Ans: We added error bars in the figure. So the data point at 20% fiber content can be considered as the apex. The case is the same for the data point at 40% fiber content.
Figure 11 b): the dP show opposite tendency vs. IFT for commercial sample and Kapok fibers – a very interesting behavior. Any idea why this was observed? What is the origin of this tendency? The Figure 3 shows an increase of the CA values for both media, so what else can affect the process?
Ans: This is a very intriguing question. Since saturation was a very complex process, droplet capturing, coalescence, drainage, and re-entrainment happened in an equilibrium state. The hollow kapok fiber owned a very different structure compared to solid PBT fiber, which could cause the different tendency. To be honest, we think it’s a complex question which needed further study. We hope in the future we can have new findings in this topic.
The English in the manuscript is not very good. There are many sytnax, grammar mistakes, which in some sections make the paper difficult to understand. A careful review, preferably by a native English speaker, is strongly recommended.
Ans: Thank you for your suggestion. We have the English in this manuscript to be checked by a native English speaker. We hope the English expression is sufficient for publication in this journal.

Reviewer 2 Report
The manuscript titled "The diesel/water separation performance of filter media containing kapok fiber" by Song et al. deals with preparation, characterization and testing of a novel filter media containing kapok fibers. The authors manufactured and tested several different types of filters, and compared one of the most promising one to a commercial filter. While the results point to a satisfactory conclusion, in my opinion the manuscript should be significantly improved before being accepted for publication. The comments are divided into major issues and smaller, easier to address comments.
Major comments:
Lines 55-56: Kapok fibers should be thoroughly discussed in the introduction, as opposed to one sentence in the current version. Origin, properties, applications etc.
Lines 56-57: The authors should provide more than one reference for application of kapok fibers, if they claim that it was often used in oil/water separation.
Line 66: It is not true that kapok fiber was not reported for diesel/water separation. See for example: Performance and mechanism of a hydrophobic-oleophilic kapok filter for oil/water separation or Evaluation of kapok (Ceiba pentandra (L.) Gaertn.) as a natural hollow hydrophobic-oleophilic fibrous sorbent for oil spill cleanup.
Lines 76-157 (Materials and methods): In general, there is lack of information about the number of parallel measurements performed. Since there are no error bars or standard deviation values anywhere, I assume that only one measurement was performed. While for some measurements, such as contact angle or filter media properties it could be looked over, it is not acceptable to describe trends in data based on single measurement. This concerns IFT measurement, drop size distribution, repeatability of filter media preparation, water content measurements and filtration tests.
Lines 130-133: How repeatable was the emulsion preparation method?
Lines 138-145: There is no information about the volume of emulsion.
General comment to Results and Discussion part: The authors do not really provide any discussion to the presented results. There is lack of explanation of the data and very few references to other literature.
Lines 195-208: Why air permeability decreases if the maximum and mean pore size in general increase? Why thickness of sample #4 is ca. 50% higher than in other samples?
Lines 221-225: Drop size is only one factor affecting the droplet stability. More pronounced effect, in my opinion, comes from the presence of a surfactant.
Lines 222-224: The presence of surfactant also leads to higher deformability of droplets, which is in contrast to what the authors wrote.
Lines 252-254: The "total volume of fibers" also increases with each incremental addition of kapok fibers, therefore it does not explain the dip in separation efficiency at 40% or higher of kapok fiber share.
Lines 258-260: There is no information about basic weight of other samples. They should be included in Table 3. How is it calculated/measured?
Lines 274-282: These are not really conclusions, just repeating abstract and values from the results section. Please rewrite this section.
Minor comments:
General advice - please check whether the use of present and past form in the manuscript is correct. For example, past form should not be used in the introduction (lines 25-43)
Line 33: ULSD is only explained in the abstract, not in the main body of the manuscript.
Line 39: "Adsorbed", not "absorbed"
Lines 39-43: To the best of my knowledge, many cars have a fuel filter before the fuel pump, therefore the shear forces occuring in the pump are irrelevant.
Line 57: I think it should state "adsorption" - fibers do not really absorb the oil.
Line 152: What is the precision of the pressure gauge?
Line 172 (Figure 3): "Size distribution" is not presented in this figure, as the caption states. The size of the water droplet should be added to caption. Why does the droplet on PBT fibers at 17 mN/m have a completely different shape that in other cases?
Line 176-179: Please rephrase this part - it is the surfactant that changes the interfacial behaviour of the drop, not the IFT.
Line 193 (Table 3): What is the percentage of kapok fiber: weight or volume?
Lines 203-204: The densities of kapok and glass fibers have wrong units.
Line 229 (Figure 8): I recommend to use log scale of the x axis. In addition, please explain what is happening in the size ranges between 0 and 5 µm.
Line 263 (Table 4): Adding the data on the other sample (#2) would make it easier to directly compare the properties of both filter media.
Author Response
The manuscript titled "The diesel/water separation performance of filter media containing kapok fiber" by Song et al. deals with preparation, characterization and testing of a novel filter media containing kapok fibers. The authors manufactured and tested several different types of filters, and compared one of the most promising one to a commercial filter. While the results point to a satisfactory conclusion, in my opinion the manuscript should be significantly improved before being accepted for publication. The comments are divided into major issues and smaller, easier to address comments.
Ans: Thanks for the comments from the reviewer. We have conducted extensive modification of this manuscript. We believe that the manuscript has been significantly improved.
Major comments:
Lines 55-56: Kapok fibers should be thoroughly discussed in the introduction, as opposed to one sentence in the current version. Origin, properties, applications etc.
Ans: In reference [13], [14] and [15], there were detailed discussion about the properties of kapok fiber. With these references, we don’t think this manuscript should repeat the discussion of kapok fibers.
Lines 56-57: The authors should provide more than one reference for application of kapok fibers, if they claim that it was often used in oil/water separation.
Ans: We added more references to claim that it was often used in oil/water separation.
Line 66: It is not true that kapok fiber was not reported for diesel/water separation. See for example: Performance and mechanism of a hydrophobic-oleophilic kapok filter for oil/water separation or Evaluation of kapok (Ceiba pentandra (L.) Gaertn.) as a natural hollow hydrophobic-oleophilic fibrous sorbent for oil spill cleanup.
Ans: Thanks for the comment. The example provided by the reviewer was about the separation of oil in water. This paper studied the separation of water in oil, which was a different topic.
Lines 76-157 (Materials and methods): In general, there is lack of information about the number of parallel measurements performed. Since there are no error bars or standard deviation values anywhere, I assume that only one measurement was performed. While for some measurements, such as contact angle or filter media properties it could be looked over, it is not acceptable to describe trends in data based on single measurement. This concerns IFT measurement, drop size distribution, repeatability of filter media preparation, water content measurements and filtration tests.
Ans: In the materials and methods, we added a statement “Five samples were prepared for each kind of filter medium”. The results were based on test of five samples for each medium. We have added error bars to the figures.
Lines 130-133: How repeatable was the emulsion preparation method?
Ans: The emulsion was very stable according to the observation in the experiments. Since it was prepared according to the SAE J1488 standard, we believed that the stability of the emulsion was not an issue for experiment.
Lines 138-145: There is no information about the volume of emulsion.
Ans: The volume of emulsion did not affect the test. It was the concentration of emulsions that mattered. For the concentration of emulsions, it was 2500 ppm by volume as mentioned in the manuscript.
General comment to Results and Discussion part: The authors do not really provide any discussion to the presented results. There is lack of explanation of the data and very few references to other literature.
Lines 195-208: Why air permeability decreases if the maximum and mean pore size in general increase? Why thickness of sample #4 is ca. 50% higher than in other samples?
Ans: Thanks for the comment of the reviewer. Air permeability was affected by fibrous structure and thickness. Although maximum and mean pore size in general increased, the media thickness decreased sharply, which led to decrease of air permeability. In theory, the thickness of sample should increase with higher kapok fiber content. We have not figured out why sample #4 was thickest, not sample #5. Our current hypothesis was due to paper formation process when kapok fiber content reached 60% or higher.
Lines 221-225: Drop size is only one factor affecting the droplet stability. More pronounced effect, in my opinion, comes from the presence of a surfactant.
Ans: We think droplet size and surfactant can both affect the droplet stability. In SAE J1488 standard, droplet size and IFT were mentioned. The two factors should both be considered.
Lines 222-224: The presence of surfactant also leads to higher deformability of droplets, which is in contrast to what the authors wrote.
Ans: As above mentioned, both droplet size and surfactant can both affect the droplet properties, including deformability. The statement emphasized the effect of droplet size since the impact of kapok fiber content on separation performance was studied under same IFT.
Lines 252-254: The "total volume of fibers" also increases with each incremental addition of kapok fibers, therefore it does not explain the dip in separation efficiency at 40% or higher of kapok fiber share.
Ans: We have revised the explanation.
Lines 258-260: There is no information about basic weight of other samples. They should be included in Table 3. How is it calculated/measured?
Ans: The basic weight was mentioned in section 2.2. It indicated the weight of paper per unit area, which was a very common measurement for paper industry.
Lines 274-282: These are not really conclusions, just repeating abstract and values from the results section. Please rewrite this section.
Ans: We have rewritten this section. Thank for the comment.
Minor comments:
General advice - please check whether the use of present and past form in the manuscript is correct. For example, past form should not be used in the introduction (lines 25-43)
Line 33: ULSD is only explained in the abstract, not in the main body of the manuscript.
Ans: We added the explanation of ULSD in the text when it appeared for the first time.
Line 39: "Adsorbed", not "absorbed"
Ans: Thanks for the comment. The error was corrected.
Lines 39-43: To the best of my knowledge, many cars have a fuel filter before the fuel pump, therefore the shear forces occuring in the pump are irrelevant.
Ans: For HPCR system, the fuel is pumped to the rail and part of fuel returns to the fuel tank. Since fuel circulates in the system, the shear force is relevant no matter the fuel filter is before or after the pump.
Line 57: I think it should state "adsorption" - fibers do not really absorb the oil.
Ans: Thanks for the comment. The error was corrected.
Line 152: What is the precision of the pressure gauge?
Ans: The max pressure can be tested was 25 kPa with precision of 0.5%.
Line 172 (Figure 3): "Size distribution" is not presented in this figure, as the caption states. The size of the water droplet should be added to caption. Why does the droplet on PBT fibers at 17 mN/m have a completely different shape that in other cases?
Ans: The previous caption was incorrect. We have corrected the caption of figure 3. From our perspective, the droplet shape size was close to others. The contact angle result at 17 mN/m was similar to that of 13 mN/m, which was reasonable.
Line 176-179: Please rephrase this part - it is the surfactant that changes the interfacial behaviour of the drop, not the IFT.
Ans: We have rephrase the expression to point out it is the surfactant that changes the interfacial behaviour of the droplet.
Line 193 (Table 3): What is the percentage of kapok fiber: weight or volume?
Ans: It was added in the table that the percentage was by weight.
Lines 203-204: The densities of kapok and glass fibers have wrong units.
Ans: Sorry for the mistake. We have corrected the errors.
Line 229 (Figure 8): I recommend to use log scale of the x axis. In addition, please explain what is happening in the size ranges between 0 and 5 µm.
Ans: The x axis was changed to log scale. The concentration droplet in the size ranges from 0 to 5 µm was pretty low, as demonstrated by Figure 8. We don’t think more explanation was needed for 0 to 5 µm droplet.
Line 263 (Table 4): Adding the data on the other sample (#2) would make it easier to directly compare the properties of both filter media.
Ans: The data of sample#2 were added in Table 4.

Reviewer 3 Report
Dear authors:
After reading your work, I have some comments to make. Firstly, let coment on the generals. It needs an in-depth revision of the grammar and writing of the paper. Please, consider making sentences shorter and more precise, where the concepts are explained correctly. Moreover, when an experimental value is given, it is necessary apport de standard deviation (it is missing in all results). Also, when talking about fiber contents, it is not correct to use only the symbol %, it should specify if that % is in volume or weight (vt or wt).
What do you mean by filter paper? Maybe, “membrane” is more correct.
What about the reproducibility of the filter and the results obtained?
Below are the specific comments for each part of the paper.
Introduction: I recommend reviewing and arguing with data the statements made, not just by citing the paper after a sentence. For example, give the values obtained in the works mentioned, to later compare with those obtained in this.
Materials and Methods: This section is not well organized. Materials need to be better explained. Before introducing the “Characterization section” should describe the fabrication: emulsion, filter, etc. The results should all be given in the results section. Please explain better all sub-section. In particular, 2.6. Diesel/water separation performance Equations (1) and (2).
Are the emulsions stable, for how long? What is the concentration of the emulsions in terms of wt.% water-diesel?
Results and discussion: This section is not well organized as a consequence of the previous section. The explanation in all sub-section needs to be improved. It would be appropriate to compare the results obtained in this work with those in the introduction section referenced.
Can you provide surface area measurements (BET)? What are the commercial samples? (reference please) Can you provide information on filter morphology after use?
Author Response
After reading your work, I have some comments to make. Firstly, let coment on the generals. It needs an in-depth revision of the grammar and writing of the paper. Please, consider making sentences shorter and more precise, where the concepts are explained correctly. Moreover, when an experimental value is given, it is necessary apport de standard deviation (it is missing in all results). Also, when talking about fiber contents, it is not correct to use only the symbol %, it should specify if that % is in volume or weight (vt or wt).
Ans: Thanks for your comment. We have conducted in-depth revision of the grammar and writing of the manuscript. We also added error bars for the experimental values and w.t. for the symbol %.
What do you mean by filter paper? Maybe, “membrane” is more correct.
Ans: Filter paper is the material made by papermaking process for filtration application. We think it is more appropriate to use “filter paper” instead of “membrane”.
What about the reproducibility of the filter and the results obtained?
Ans: In section 2.3, it was added that “Five samples were prepared for each kind of filter medium”. The test results were the average value of five samples. Error bars were provided in the figures.
Below are the specific comments for each part of the paper.
Introduction: I recommend reviewing and arguing with data the statements made, not just by citing the paper after a sentence. For example, give the values obtained in the works mentioned, to later compare with those obtained in this.
Ans: We think the statements were important for citing the reference, since the values in the works depended on specific testing conditions and were not comparable among different studied. However, the trend of data and conclusions were useful for the filter medium study.
Materials and Methods: This section is not well organized. Materials need to be better explained. Before introducing the “Characterization section” should describe the fabrication: emulsion, filter, etc. The results should all be given in the results section. Please explain better all sub-section. In particular, 2.6. Diesel/water separation performance Equations (1) and (2).
Ans: We have reorganized this section to make it more clear to understand. The sub-sections were (1) Fibers and chemicals, (2) Filter media preparation, (3) Characterization of fibers and filter media and (4) Diesel/water separation performance. We also explained the equations (1) and (2) which we think it’s straightforward to understand.
Are the emulsions stable, for how long? What is the concentration of the emulsions in terms of wt.% water-diesel?
Ans: The emulsion was very stable according to the observation in the experiments. Since it was prepared according to the SAE J1488 standard, we believed that the stability of the emulsion was not an essential issue for experiment. For the concentration of emulsions, it was 2500 ppm by volume as mentioned in the manuscript. We choose to use concentration by volume instead of concentration by weight.
Results and discussion: This section is not well organized as a consequence of the previous section. The explanation in all sub-section needs to be improved. It would be appropriate to compare the results obtained in this work with those in the introduction section referenced.
Ans: We changed the titles of sub-sections to make it more clear to understand. We believed the framework of this section can demonstrate the results and discussion orderly. As to compare the results obtained in this work with those in the introduction section referenced, the data were not comparable for different test condition and test bench. It’s more useful to present the trend of results and draw conclusions based on the results from the same test.
Can you provide surface area measurements (BET)? What are the commercial samples? (reference please) Can you provide information on filter morphology after use?
Ans: Since the diameter of kapok fiber was micro-sized, we don’t think it’s useful to measure the surface area. The changes of kapok fiber slightly affected the surface area, which provided limited information. The commercial samples were provided from the major manufacturer in the market. To our best knowledge, there was no reference directly related to this sample. It was mentioned in section 2.1 that commercial filter samples were composed by polybutylene terephthalate (PBT) fiber layer and cellulose fiber layer. For the filter morphology after use, we will further study this issue and provide more information on filter morphology in future publications.

Round 2
Reviewer 2 Report
In general, I am satisfied with the response from the authors. However, there are few things that I would ask them to re-consider (mostly from my original review).
"Line 66: It is not true that kapok fiber was not reported for diesel/water separation. See for example: Performance and mechanism of a hydrophobic-oleophilic kapok filter for oil/water separation or Evaluation of kapok (Ceiba pentandra (L.) Gaertn.) as a natural hollow hydrophobic-oleophilic fibrous sorbent for oil spill cleanup.
Ans: Thanks for the comment. The example provided by the reviewer was about the separation of oil in water. This paper studied the separation of water in oil, which was a different topic."
In that case, please specify in the text and title that the topic is separation of water from water-in-Diesel emulsions. Diesel/water separation is a generic term and can mean both oil-in-water and water-in-oil treatment.
"Line 229 (Figure 8): I recommend to use log scale of the x axis. In addition, please explain what is happening in the size ranges between 0 and 5 µm.
Ans: The x axis was changed to log scale. The concentration droplet in the size ranges from 0 to 5 µm was pretty low, as demonstrated by Figure 8. We don’t think more explanation was needed for 0 to 5 µm droplet."
I respectfully disagree. By volume, it might be insignificant, however one could observe a growing "bump" coming from the smallest droplets. This can be both the result of lower interfacial tension and creating satellite drops during emulsification or effect of multiscattering.
Author Response
In general, I am satisfied with the response from the authors. However, there are few things that I would ask them to re-consider (mostly from my original review).
(1) "Line 66: It is not true that kapok fiber was not reported for diesel/water separation. See for example: Performance and mechanism of a hydrophobic-oleophilic kapok filter for oil/water separation or Evaluation of kapok (Ceiba pentandra (L.) Gaertn.) as a natural hollow hydrophobic-oleophilic fibrous sorbent for oil spill cleanup.
Ans (round 1): Thanks for the comment. The example provided by the reviewer was about the separation of oil in water. This paper studied the separation of water in oil, which was a different topic."
(Round 2) In that case, please specify in the text and title that the topic is separation of water from water-in-Diesel emulsions. Diesel/water separation is a generic term and can mean both oil-in-water and water-in-oil treatment.
Ans (round 2): Thanks for your comment. We have revised the title to “Separation of Water in Diesel Using Filter Media Containing Kapok Fibers”. In line 12, “diesel/water separation” was changed to “water separation in diesel”. In line 15, “diesel/water separation” was changed to “water separation”.
(2) "Line 229 (Figure 8): I recommend to use log scale of the x axis. In addition, please explain what is happening in the size ranges between 0 and 5 µm.
Ans (round 1): The x axis was changed to log scale. The concentration droplet in the size ranges from 0 to 5 µm was pretty low, as demonstrated by Figure 8. We don’t think more explanation was needed for 0 to 5 µm droplet."
(Round 2) I respectfully disagree. By volume, it might be insignificant, however one could observe a growing "bump" coming from the smallest droplets. This can be both the result of lower interfacial tension and creating satellite drops during emulsification or effect of multiscattering.
Ans (round 2): We think that the water droplets were secondarily emulsified by forming satellite drops when the emulsion was pumped through the pipes (the inner diameter of 6 mm) and valves (the inner diameter of 3 mm) [1], causing a growing "bump" when water droplets were less than 5 µm.
[1] X. Zhang, O.A. Basaran, An experimental study of dynamics of drop formation, Physics of Fluids, 7 (1995) 1184-1203.

Reviewer 3 Report
Dear authors:
Thanks for your answers.
I consider that after the revision, the paper has gained in clarity. Still, I have a few concerns.
Can you provide the amount of monoolein or clay to achieve different IFT used? It is essential because also you mentioned in Pg. 8 line 215 “As shown in Figure 7, IFT decreased with the increase of monoolein”
Pg. 3 line 99, Table 1, errors are missing
Pg. 5 line 169, The image for 132⁰, I believe that it is deformed.
Pg. 7 line 195, Table 3, errors are missing
Pg.10 line 266 Table 4, errors are missing
Have a nice day
Author Response
I consider that after the revision, the paper has gained in clarity. Still, I have a few concerns.
(1) Can you provide the amount of monoolein or clay to achieve different IFT used? It is essential because also you mentioned in Pg. 8 line 215 “As shown in Figure 7, IFT decreased with the increase of monoolein”.
Ans: Monoolein was added to achieve concentration of 115 ppm, 180 ppm and 263 ppm by volume in the emulsion, respectively, which can lead to IFT of 21 mN/m, 17 mN/m and 13 mN/m. This sentence was added in line 215 to 217.
Pg. 3 line 99, Table 1, errors are missing
Ans: Since all the fibers were poly-dispersed, we showed the range of fiber diameter rather than adding errors to the average diameter. So we did not modify Table 1.
Pg. 5 line 169, The image for 132⁰, I believe that it is deformed.
Ans: In the manuscript, we paid more attention to the value contact angle. We will conduct more study on the details of water droplet deformation.
Pg. 7 line 195, Table 3, errors are missing
Ans: The errors of mean pore size, maximum pore size, thickness and air permeability were added in Table 3 (line 195).
Pg.10 line 266 Table 4, errors are missing
Ans: The errors of basic weight, thickness, air permeability, mean pore size and maximum pore size were added in table 4 (line 267).
